# Strategic Successive Harvesting of Rocket and Spinach Baby Leaves Enhanced Their Quality and Production Efficiency

Filippos Bantis *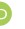, Chrysos Kaponas, Charalambos Charalambous and Athanasios Koukounaras

Department of Horticulture, Aristotle University, 54124 Thessaloniki, Greece; ckaponas@agro.auth.gr (C.K.); charchar@agro.auth.gr (C.C.); thankou@agro.auth.gr (A.K.)
* Correspondence: fbantis@agro.auth.gr; Tel.: +30-2310-994123

**Abstract:** Rocket and spinach baby leaves are valuable commodities since they are basic components of popular ready-made salads. Two methods may follow after harvesting: establishment of new cultivations or successive revegetations and harvests. This study aimed to investigate the yield and nutritional value of rocket and spinach baby leaves after individual cultivations or successive revegetations in a floating system to improve their production strategy. The crops were cultivated in a greenhouse for seven weeks using a floating system with an adjusted nutrient solution. The leaves were either harvested and immediately replaced with a new set of plants (control) or harvested and placed again in the same tank in order to revegetate (revegetation). Revegetated rocket baby leaves in five cuts produced similar yield, with greater antioxidant capacity (DPPH scavenging activity) and total phenolic content, and greater nitrate content (eight times below the maximum allowed by EU) compared to control. Revegetated spinach produced more yield with enhanced antioxidant activity and total phenolic content and the same nitrate content compared to the control. Colour was not affected in either crop, thus eliminating the possibility for market rejection. Production efficiency was increased, as shown by the yields and the reduced resources provided in the revegetation tank. Thus, successive harvesting and revegetation are suggested for increased production efficiency and quality of rocket and spinach baby leaves.

**Keywords:** revegetation; *Eruca sativa*; *Spinacia oleracea*; floating system; antioxidant capacity; nitrate content

## 1. Introduction

Nowadays, baby leaf vegetables such as spinach, rocket, and lettuce are economically important products since they are basic components of ready-made salads. Fresh-cut vegetables' marketability shows increasing popularity throughout the world due to the species variability and easiness to handle by producers and consumers [1]. An essential attribute for baby leaf vegetables is their nutritional value which is known to be greater compared to the fully grown plants [2]. Studies have shown that vegetable consumption reduces the possibility of cardiovascular diseases and some cancer types due to antioxidant compounds (i.e., phenolics and vitamins, among others), which show protective roles [3].

Rocket (*Eruca sativa* Mill.) is a popular leafy vegetable due to its distinct taste and nutritional content. Specifically, rocket leaves are rich in vitamins such as A, B, C, and K, iron, and essential proteins which contribute to human health [2]. Spinach (*Spinacia oleracea* L.) is another important leafy vegetable with high nutritional content, as it is classified third in antioxidant capacity among vegetables [4]. The crop is widely cultivated and consumed since it is rich in vitamins A, B, and C, polyunsaturated fatty acids, iron, and magnesium [5]. Both crops are ideal for baby leaf production, as they have high nutritional content and exert significant antioxidant activity [4,6]. Both commodities have a rich green colour, which is their main qualitative characteristic as perceived by consumers [7].

Apart from the beneficial nutritional qualities mentioned above, leafy vegetables contain high amounts of nitrates which impose harmful effects in humans as they are

metabolized to nitrites and nitrosamines. Vegetables are the main source of nitrate consumption in the European region [8], and thus, the European Union passed legislation for the maximum allowed nitrate amounts in certain leafy vegetables. Specifically, according to the European Commission Regulation No 1258/2011, the maximum allowed nitrate amounts for fresh spinach is 3500 mg $NO_3$/kg, while the maximum for rocket is 6000 or 7000 mg $NO_3$/kg depending on the time of harvest (April to September and October to March, respectively). Among all vegetables, rocket contains the highest amount of nitrates [9]. Nitrate amounts up to 9300 mg/kg have been reported for rocket grown in Italy [10].

Among several methods of hydroponically grown crops, the floating system is the easiest and cheapest means of leafy vegetables' production [11]. Floating is a closed system with important benefits such as low cost, water and nutrient use efficiency, environmentally friendly, production efficient, and it provides the option for mechanization and automation of crop production [12,13]. Moreover, soil-borne diseases and pests are completely avoided [14,15], while the number of cultivation cycles within a year is increased. Nowadays, baby leaf vegetables are widely produced using floating systems in plant factories with artificial lighting (PFAL), where their production and resource use efficiency is enhanced. Rocket and spinach have already been tested in PFAL systems showing promising potential for increased yield and nutritional quality [16,17].

Successive harvesting is a widely practiced agronomic method for the regulation of leafy vegetables' bioactive compound accumulation. In this method, physiological adjustments occur that subsequently lead to altered plant metabolism [18]. Both spinach and rocket are harvested by cutting them at about 3 cm above the ground, 20 to 60 days after transplanting, depending on the season of cultivation, environmental conditions, and market needs. These leafy vegetables revegetate after the first harvest, and thus, it is possible to conduct four or five successive harvests in a few weeks since their roots remain intact. Apart from time shortening between harvests, this approach results in cost reduction, resource use limitation, and labor facilitation [11]. Specifically, compared to several individual sowings, cultivations, and harvests, revegetation facilitates baby leaf production by limiting the initial resources needed (i.e., seeds, trays, substrates), as well as conserving water and nutrients, which are only replenished when needed. This is particularly important for baby leaves' cultivation in soil, where successive harvests might facilitate their overall production. Moreover, this technique allows better time organization, which can also be implemented for hydroponic cultivation in modern PFAL systems, where different light wavelengths might enhance baby leaves' production and quality. Production efficiency reaches its highest limit when plant production can no longer increase without sacrificing another aspect of plant production. In a study with rocket, the authors suggested that more than two successive harvests are not profitable for soil cultivation [19], which needs significantly more attention than the floating system. However, the literature lacks information about the strategic revegetation of rocket and spinach in a floating system with a view to enhanced total yield and bioactive compounds. Therefore, the objective of the present study was to test the overall yield and nutritional value of rocket and spinach baby leaves after individual cultivations or after successive revegetations and harvests in order to improve the strategy of their production.

## 2. Materials and Methods

### 2.1. Plant Material and Experimental Design

The experiment was conducted in a glass greenhouse at the Laboratory of Vegetable Crops, Aristotle University of Thessaloniki, Greece. Rocket (*Eruca sativa* Mill.) and spinach (*Spinacia oleracea* L.) seeds were sown in 128-cell polystyrene plug trays (33 × 66 cm, 588 plants/m²) filled with enriched peat. Upon the first true leaf formation, trays were thinned up to one plant per cell, and afterwards, they were placed in two metallic tanks (2 m length × 0.8 m width × 0.24 m height) per crop, filled with 200 L Hoagland solution (100% strength) each [20]. In both tanks of each crop, pH was constantly adjusted to

$6.8 \pm 0.3$, electrical conductivity (EC) was constantly adjusted to $2.8 \pm 0.3$ mS/cm, while water temperature was fluctuating between 6 and 15 °C. Rocket and lettuce baby leaves were reported to have optimum production at 20 °C root zone temperature [21]. Therefore, 6–15 °C falls within the favorable root zone temperatures, while they are typical for the experiment's period.

Plants were harvested at the baby leaf stage (about 10 cm leaf length). The leaves were cut at 2–3 cm from the substrate according to usual harvesting practices. In the control tank (control), harvested trays were immediately replaced with trays that were sown a few days earlier to contain plants in the stage of one true leaf. In order for the control tank to be immediately used with plants of the appropriate age, three successive sowings were done two days apart. In the revegetation tank (revegetation), harvested trays were placed again in the same tank and cultivated in order to revegetate. This resulted in three and five harvests for the control and the revegetation trays, respectively, in a period of seven weeks (Table 1). Each treatment consisted of three trays-replications.

**Table 1.** Dates of harvest and days in the floating systems of rocket and spinach baby leaves cultivated in control or revegetation tanks. Days from sowing to first harvest are shown in brackets.

| Tank | Date of Harvest | | | | |
|---|---|---|---|---|---|
| | 16/1 | 7/2 | 17/2 | 27/2 | 7/3 |
| **Control** | | | | | |
| Harvest | First | - | First | - | First |
| Growth cycle | 34 (46) days | - | 32 (43) days | - | 18 (28 days) |
| **Revegetation** | | | | | |
| Harvest | First | Second | Third | Fourth | Fifth |
| Growth cycle | 34 (46) days | 22 days | 10 days | 10 days | 8 days |

### 2.2. Experimental Determinations

During cultivation, growing conditions such as pH, EC, temperature, and nutrient solution level were frequently recorded. Upon each harvest, leaf mass per area (i.e., yield) was measured, and plants were stored at −20 °C until their nutritional content was determined. A digital colorimeter (CR-400 Chroma Meter, Konica Minolta Inc., Tokyo, Japan) was used to determine colorimetric parameters such as lightness (0–100: black to white), chroma (colour saturation), and hue angle (redness to greenness), which were measured on the leaf tip, in leaves of 20 plants [22].

In order to determine the nutritional content, 100 g of plants from each tray were mashed into a pulp. Soluble sugar content was measured with a refractometer (PAL-α, Atago, Tokyo, Japan) after filtering part of the pulp.

Nitrate content was determined chromatographically, according to Cataldo et al. [23]. Briefly, 2.5 g of pulp was extracted with 25 mL $H_2O$. Aliquot 0.2 mL of the extract was pipetted to 0.8 mL $H_2SO_4$ or 0.8 mL 5% salicylic acid to $H_2SO_4$, followed by 19 mL 2N NaOH addition. The coloured product was placed in a spectrophotometer (UV-VIS, Shimadzu Scientific Instruments, Columbia, MD, USA), and its absorbance was measured at 410 nm, while results were expressed as mg/kg.

Total phenolic content was determined chromatographically according to Singleton and Rossi [24]. Briefly, 2 g of pulp was extracted with 25 mL 80% aqueous methanol. Aliquot 0.5 mL of the extract was pipetted to 2.5 mL 10% Folin–Ciocalteu and 2 mL 7.5% $Na_2CO_3$, followed by 5 min at 50 °C. The coloured product was placed in a spectrophotometer, and its absorbance was measured at 760 nm, while results were expressed as mg gallic acid equivalent/g.

Total antioxidant activity was determined chromatographically using DPPH (1,1-diphenyl-2-picryl hydrazyl) method, according to Brand-Williams et al. [25]. Briefly, 2 g of pulp was extracted with 25 mL 80% aqueous methanol. Aliquot 50 μL of the extract was pipetted to 2950 μL of 100 μM DPPH in methanol and stored in darkness. Exactly

30 min later, the coloured product was placed in a spectrophotometer; its absorbance was measured at 517 nm, while results were expressed as mg ascorbic acid equivalent/100 g.

Statistical analysis was performed using SPSS software (SPSS 25.0, IBM Corp., Armonk, NY, USA). Analysis of variances (ANOVA) and Tukey post-hoc tests were conducted at a significance level $p = 0.05$.

## 3. Results and Discussion

The floating system is ideal for the cultivation of baby leaf vegetables due to its high productivity, water and nutrient use efficiency, and green nature [13]. In general, both rocket and spinach grew normally without biotic and abiotic disorders and formed uniform leaves. At the end of the experimental period, the total yield of rocket was similar in both tanks (Figure 1A). However, spinach baby leaves had a significantly greater total yield in the revegetation tank compared to the control with 53% higher values (Figure 1B). This is particularly important since the revegetation tank reduced the production cost due to minimum sowings, less water and nutrient provision, and limited person-hours, thus leading to higher production with much lower inputs. In addition, the reduction of inputs is nowadays a desirable trait by consumers since it is critical for environmental sustainability [26]. As per individual harvests of rocket, the second cut was significantly more productive in the control tank compared to the first and third cuts, while the first cut was the most productive in the revegetation tank compared only to the fourth cut (Figure 1C). In spinach, the second cut led to greater leaf mass per area compared to the first and third cuts in the control, while the second cut was the most productive compared to all the other cuts in the revegetation tank (Figure 1D). A yield reduction was somehow expected in successive cuts compared to the first cut even before the experiment started. However, the revegetation tank needed substantially less time between successive cuts, thus leading to equal (rocket) or significantly more leaf mass per area (spinach) compared to individual sowing and harvests (control). On the control tank, baby leaves were cultivated for about 34 days before harvesting the first set of trays, followed by 32 and 18 days of cultivation in the floating system before harvesting the second and third set of trays, respectively. In the revegetation tank, baby leaves were cultivated for 34 days before the first cut, 22 more days for the second cut, and only 8 to 10 more days for the third, fourth, and fifth cuts. In another study with rocket, revegetated plants decelerated their root growth due to the absence of leaves, which provide the most photosynthates [27]. This is certainly the reason why successive harvests showed a yield-decreasing tendency in the rocket of our study. The same tendency appeared in spinach after the second cut. However, revegetated spinach already had a well-developed root system which boosted the overground development after each cut, leading to a higher total yield. Similar to spinach, in a study with three green Genovese basil cultivars, plants produced significantly more yield in the second cut compared to the first cut [28]. On the contrary, green and red lettuce exhibited greater yield in leaves harvested at the first cut compared to the second cut [18], while Corrado et al. [29] found greater harvest index (i.e., leaf mass-to-total biomass), as well as leaf weight and leaf area in the first cut of sweet basil compared to the second cut.

Colour is the most important visual characteristic which defines the consumer's preferences [30]. Especially for rocket and spinach, deep green colour is associated with healthy leaves, rich in bioactive compounds, minerals, and high nutritional content. In our case, colorimetric parameters such as lightness, chroma, and hue angle did not show any significant differences in either species (Table 2). The fact that colour was virtually unchanged between the first harvest and harvests from successive revegetations is critical since the visual quality remained stable and thus prohibiting the markets from avoiding the product. Similarly, green Genovese basil did not show significant colour differences between the first and second cut [28]. According to Madeira et al. [31], colour lightness is strongly correlated with chlorophyll and carotenoid concentration.

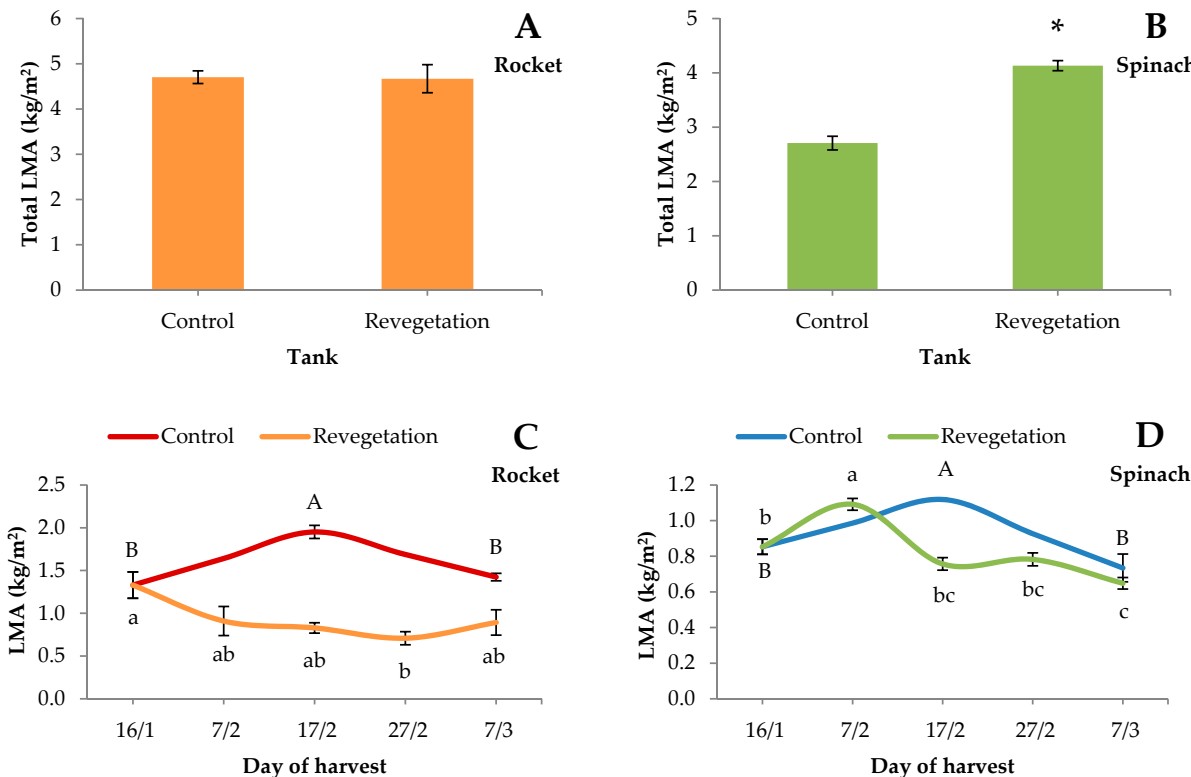

**Figure 1.** Total leaf mass per area (LMA) (**A**,**B**) and individual LMA (**C**,**D**) of rocket (**A**,**C**) and spinach (**B**,**D**) baby leaves cultivated in control or revegetation tanks. Means of control and revegetation tanks were computed from three and five harvests, respectively. In (**A**,**B**), bars (±SE) followed by an asterisk are significantly greater ($p \leq 0.05$). In (**C**,**D**), lines (±SE) followed by different letters (upper-case or lower-case) are significantly different ($p \leq 0.05$).

**Table 2.** Colorimetric parameters and soluble sugar content (SSC) of rocket and spinach baby leaves cultivated in control or revegetation tanks. Means of control and revegetation tanks were computed from three and five harvests, respectively. Means (± SE) of colorimetric parameters and SSC were calculated from 20 and 3 samples, respectively.

|  | Lightness | Chroma | Hue Angle | SSC |
|---|---|---|---|---|
| **Rocket** | | | | |
| Control | 44.78 ± 0.49 | 37.86 ± 0.66 | 122.76 ± 0.28 | 3.93 ± 0.10 |
| Revegetation | 45.43 ± 0.47 | 37.51 ± 0.71 | 122.99 ± 0.33 | 4.12 ± 0.03 |
| **Spinach** | | | | |
| Control | 38.94 ± 0.21 | 20.98 ± 0.51 | 125.12 ± 0.11 | 5.86 ± 0.12 |
| Revegetation | 37.84 ± 0.36 | 19.74 ± 0.23 | 125.08 ± 0.08 | 6.53 ± 0.34 |

Soluble sugar content is a means of quantifying the taste of leafy vegetables and evaluating their nutritional quality. Even though a tendency for greater values was observed in the revegetation tank of both species, the soluble sugar content was not significantly affected by the different harvesting methodologies (Table 2). It is expected that plants from both tanks will have similar sweetness. Bell et al. [32] reported that consumers preferred rocket leaves from the first cut compared to leaves from successive cuts, and the result was associated with higher glucosinolate concentrations which increased the bitterness and peppery perceptions. Fallovo et al. [12] observed that reduced sugar content was associated with increased nitrate content which was not evident in our case.

Nitrates are widely known to be harmful to human health due to their relation with carcinogenic compounds such as nitrosamines [33]. In the human diet, vegetables such as rocket and spinach are the main source of nitrates [34]. In rocket, the commonly cultivated

crop with the highest amounts of nitrates, nitrate content was significantly higher in the revegetation tank (+8%) compared to the control (Figure 2A). However, no significant differences in nitrate accumulation were observed in spinach (Figure 2B). As per individual harvests of rocket, leaves from the first cut produced significantly more nitrates compared to the second cut in the control tank, while no differences were observed in the revegetation tank among cuts (Figure 2C). In spinach, there was a tendency for decreasing nitrate content moving from January to March, but no significant differences were observed in the nitrate content of leaves from different cuts, both in the control and the revegetation tanks (Figure 2C,D). Nitrates are known to be accumulated under low light conditions, which reduce the activity of nitrate reductase [9]. Even though the experiment was conducted between January and March when the photoperiod of natural light is about 9.30–11.30 h, and the total incident light is limited due to natural phenomena, nitrate content in both crops reached very low values. Specifically, nitrate concentration in rocket and spinach was about 8 to 10 times lower than the maximum allowed values of 7000 (harvest from October to March) and 3500 mg/kg, respectively, as stated by European Commission Regulation No 1258/2011. Rocket plants harvested twice were found to have greater nitrogen use efficiency and nitrogen uptake due to greater biomass production compared to plants harvested once [27]. In another study, the nitrate content of green and red lettuce was found greater in the second cut, while the first cut exhibited lower values [18]. Sweet basil developed more nitrates and total nitrogen in the second cut compared to the first cut [29]. Ciriello et al. [28] found that the nitrate content of three basil cultivars was variably affected by the two successive cuts and commented that a clear genotypic impact ensued. *Cichorium spinosum* leaves produced lower nitrate amounts in the first cut compared to three or two successive cuts in September and December, respectively [35].

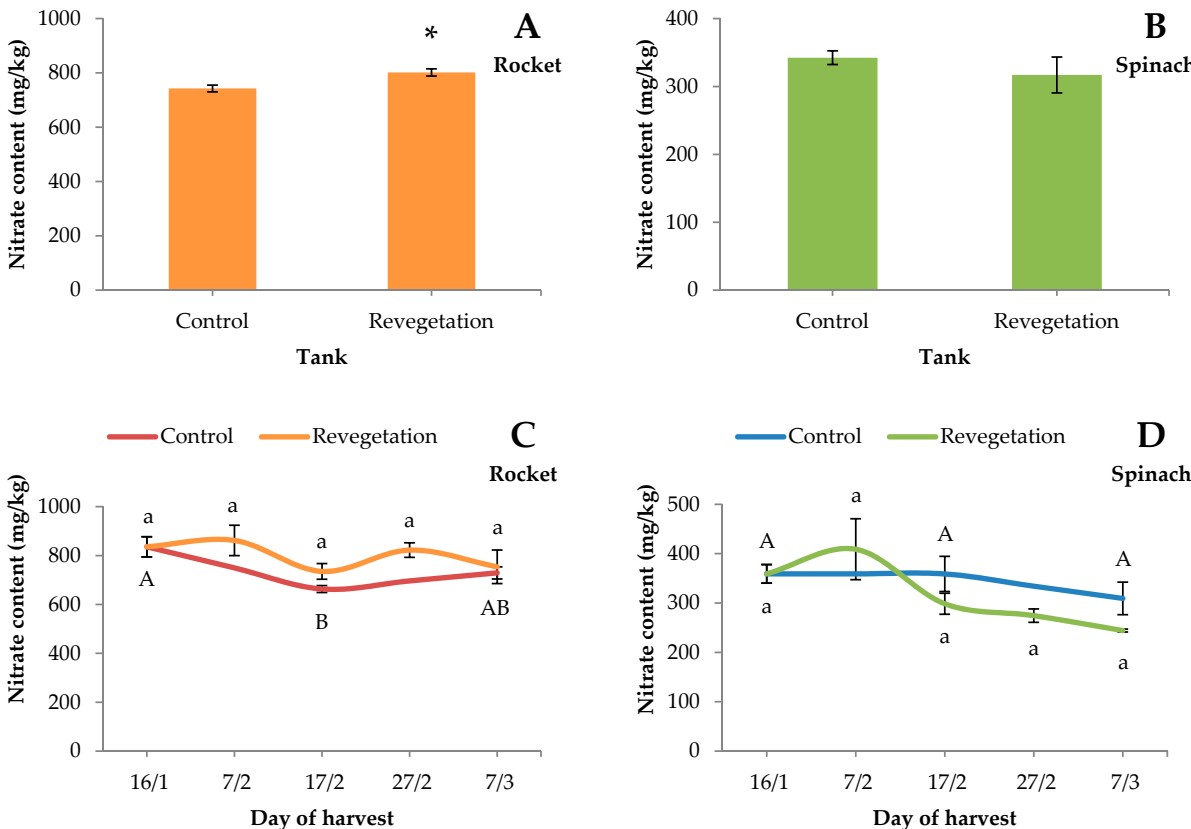

**Figure 2.** Nitrate content derived from all harvests (**A**,**B**) or from individual harvests (**C**,**D**) of rocket (**A**,**C**) and spinach (**B**,**D**) baby leaves cultivated in control or revegetation tanks. Means of control and revegetation tanks were computed from three and five harvests, respectively. In (**A**,**B**), bars (±SE) followed by an asterisk are significantly greater ($p \leq 0.05$). In (**C**,**D**), lines (±SE) followed by different letters (upper-case or lower-case) are significantly different ($p \leq 0.05$).

Plants form and accumulate bioactive compounds as a means of adaptation to environmental stresses, including drought, salinity, excess light, and diseases. Phenolic compounds are one of the most important groups of secondary metabolites which are produced in response to the abovementioned stress factors and exert several activities involving plant defense and signaling [36]. Both in rocket and spinach, total phenolic content was significantly greater in the revegetation tank compared to the control (+25 and +18%, respectively) (Figure 3A,B). No significant differences were exhibited in the control tank among individual harvests of rocket. However, rocket in the revegetation tank showed a tendency for increased total phenolic content since the first cut had significantly lower values compared to the third, fourth, and fifth cuts, while the second cut had lower values compared to the fourth cut (Figure 3C). In spinach cultivated with the control methodology, total phenolic content was significantly lower in the second cut compared to the first and third cuts. Similar to the case of rocket, spinach in the revegetation tank showed a tendency for increased total phenolic content with significantly greater values in the fifth cut compared to the first and third cuts (Figure 3D).

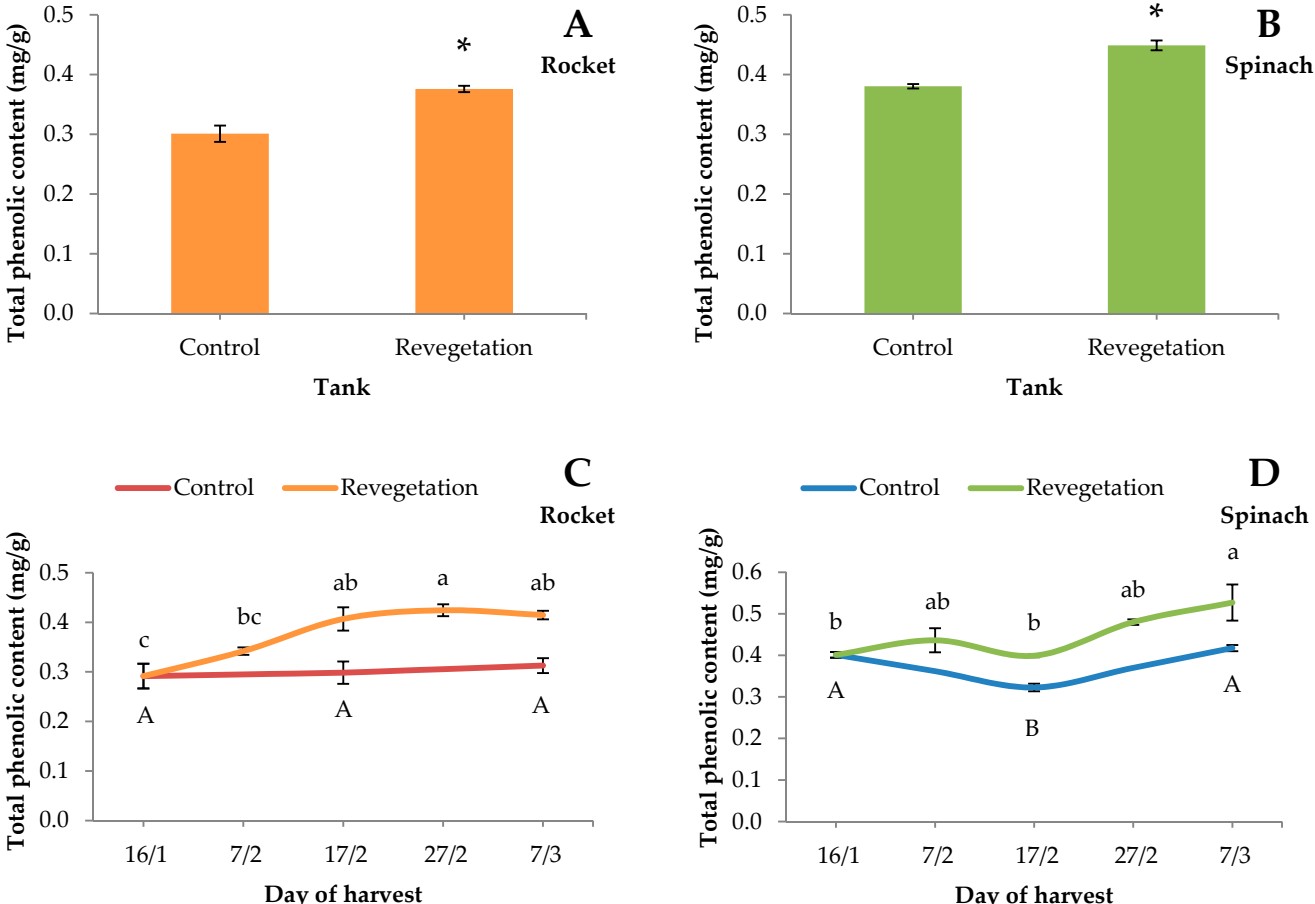

**Figure 3.** Total phenolic content derived from all harvests (**A,B**) or from individual harvests (**C,D**) of rocket (**A,C**) and spinach (**B,D**) baby leaves cultivated in control or revegetation tanks. Means of control and revegetation tanks were computed from three and five harvests, respectively. In (**A,B**), bars (±SE) followed by an asterisk are significantly greater ($p \leq 0.05$). In (**C,D**), lines (±SE) followed by different letters (upper-case or lower-case) are significantly different ($p \leq 0.05$).

Vegetables are important for a healthy and balanced diet since they provide a set of antioxidant compounds which regulate oxidative stress in humans [37]. Both in rocket and spinach, antioxidant capacity portrayed by DPPH scavenging activity was significantly greater in the revegetation tank compared to the control (+20 and +24%, respectively) (Figure 4A,B). As per individual harvests of rocket, both the control and the revegetation

tanks led to significantly greater DPPH scavenging activity in the first cut (control), and first and second cuts (revegetation), compared to the successive cuts (Figure 4C). On the contrary, spinach showed a tendency for increasing values, especially in the revegetation tank, where the fifth cut had significantly greater antioxidant capacity compared to the first cut. Spinach in the control tank displayed significantly greater DPPH scavenging activity in the third cut compared to the second cut (Figure 4D). Quite similar to the case of rocket, sweet basil exhibited greater antioxidant properties in leaves of the first cut compared to the second cut, while total polyphenols did not show significant differences [29]. Moreover, Carillo et al. [18] reported greater total phenolic content and antioxidant capacity in the second cut of lettuce leaves compared to the first cut. Petropoulos et al. [35] found greater phenolic acids, flavonoids, and total phenolic content in the first cut of *Cichorium spinosum* leaves compared to successive cuts, but the opposite was reported for DPPH scavenging activity, which was reduced in the first cut.

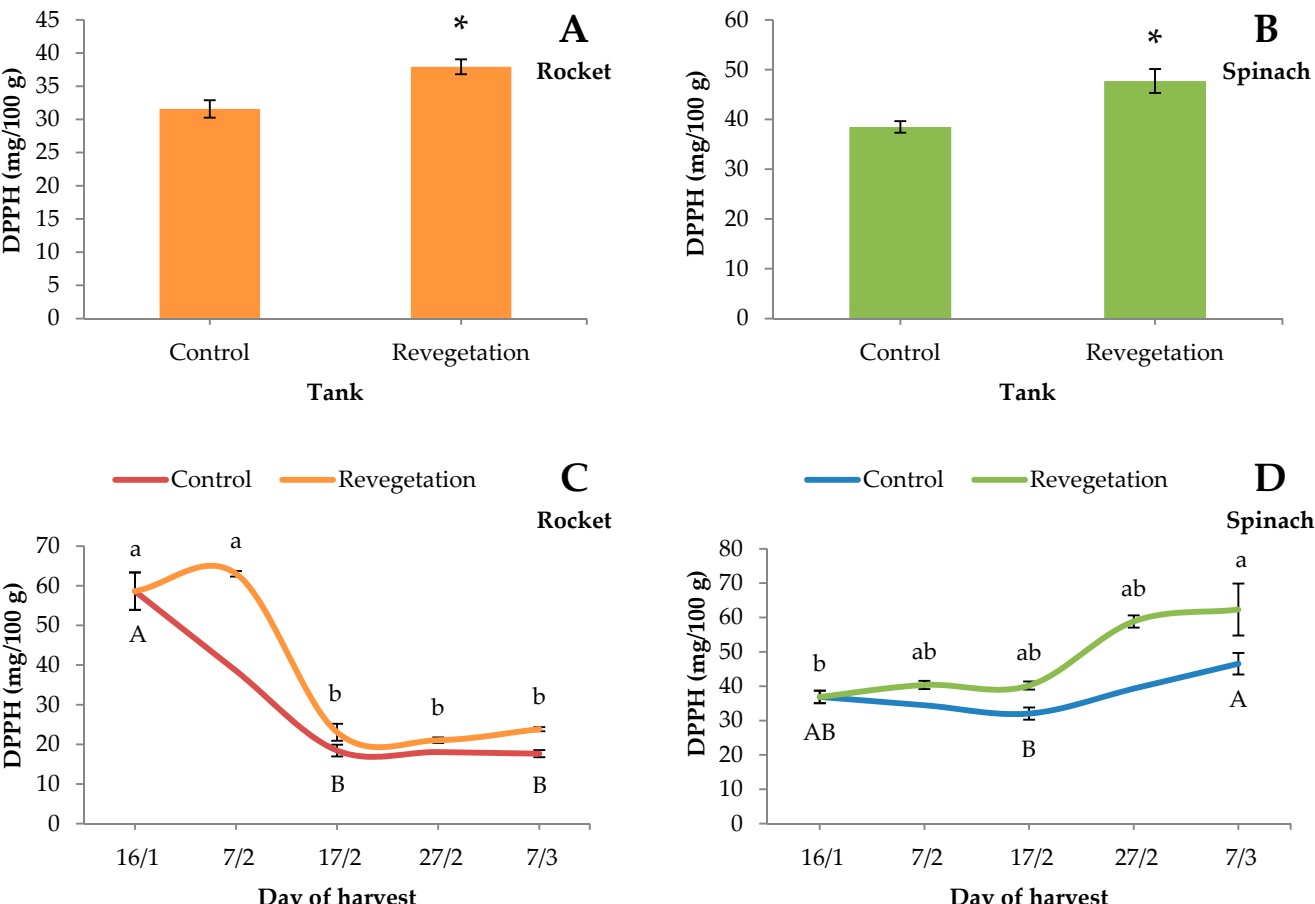

**Figure 4.** DPPH scavenging activity derived from all harvests (**A**,**B**) or from individual harvests (**C**,**D**) of rocket (**A**,**C**) and spinach (**B**,**D**) baby leaves cultivated in control or revegetation tanks. Means of control and revegetation tanks were computed from three and five harvests, respectively. In (**A**,**B**), bars (±SE) followed by an asterisk are significantly greater ($p \leq 0.05$). In (**C**,**D**), lines (±SE) followed by different letters (upper-case or lower-case) are significantly different ($p \leq 0.05$).

## 4. Conclusions

In order to facilitate the production process and increase the production efficiency of baby leaf vegetables, it is of utmost importance to form a strategy involving all necessary parameters such as cost, resources, labor, and product quality. The abovementioned parameters can be met by allowing harvested baby leaves to revegetate and proceed to further cuts. Revegetated rocket produced the same yield with enhanced antioxidant potential, including phenolics, compared to rocket cultivated in successive sowings. Revegetated

rocket's nitrate content was higher than the rocket cultivated in successive sowings but way below the maximum allowed amounts stated by the European Commission. Revegetated spinach baby leaves produced a greater total yield with enhanced antioxidant capacity, including phenolics, and the same nitrate content compared to spinach cultivated in successive sowings. Colour, the most important visual quality characteristic for consumers, was not affected in either species, thus eliminating the possibility for market rejection. It is clear that production efficiency was enhanced as demonstrated by the similar (in rocket) and even greater (in spinach) yield in the revegetation tanks and taking into consideration the reduced inputs provided in the latter treatment. Therefore, successive harvesting and revegetation are suggested for increased production efficiency and quality of rocket and spinach baby leaves.

**Author Contributions:** Conceptualization, methodology, and data analysis: F.B. and A.K.; experimental measurements: C.K. and C.C.; writing—original draft preparation: F.B.; writing—review and editing: F.B. and A.K.; supervision and project administration: A.K. All authors have read and agreed to the published version of the manuscript.

**Funding:** This research received no external funding.

**Institutional Review Board Statement:** Not applicable.

**Acknowledgments:** The authors would like to express their gratitude to Kalliopi Radoglou who critically reviewed the manuscript and commented on its English.

**Conflicts of Interest:** The authors declare no conflict of interest.

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
