# Peer review of "Strategic Successive Harvesting of Rocket and Spinach Baby Leaves Enhanced Their Quality and Production Efficiency"

_agriculture, doi:10.3390/agriculture11050465_

Round 1

Reviewer 1 Report

You mentioned productions efficiency few times (line 21, 59, 304 and 316) bur there are no results about the production efficiency.

What means aka yield (104)?

All figures should marked with rocket and spinach instead of A and C or B and D, it more clear to read

Author Response

You mentioned productions efficiency few times (line 21, 59, 304 and 316) bur there are no results about the production efficiency.

  • Response: Statements about production efficiency were included in the abstract (L22-23), the introduction (L84-85), and the conclusions’ (L337-340) sections, as suggested.

What means aka yield (104)?

  • Response, L123: “aka” was substituted with “i.e.” (in essence).

All figures should marked with rocket and spinach instead of A and C or B and D, it more clear to read

  • Response: All figures were marked with “rocket” and “spinach” as suggested.

Reviewer 2 Report

The manuscript “Strategic successive harvesting of baby leaf vegetables enhanced their quality and production efficiency” is based on a strategy of successive harvesting and revegetation of rocket and spinach baby leaves in soilless culture for increased production efficiency and quality.

To sum up, this experiment can find an interest for the specialists in this field as well as it is applicable in soilless culture system the for their economical production. The following minor revisions are suggested.

Title: The title is fine and supports the arguments which have been addressed in this research, but include the names of both leafy vegetables (preferably scientific names) in the title.

Abstract:

Abstract is fine and appropriate to sum up the whole study.

Keywords: Keywords are appropriate

Introduction

Introduction is good and well elaborated the scope of study.  

Materials and Methods

Methodology of the proposed study is fine and well designed, but it will be more convenient if provide pictures of replantation especially revegetation or describe the intensity of harvesting to revegetate. 

Results and Discussion

The results are described properly and supported with references but the minor changes are required as follow;

Line 136-146: These lines should be the part of Introduction section instead of Results and Discussions.

Line 155-160: Use the words “first, second and third etc.” instead of 1st, 2nd and 3rd etc.

Provide the pictures for each harvest in supplementary files

References

No reference is found unformatted.

Good Luck!

Author Response

The manuscript “Strategic successive harvesting of baby leaf vegetables enhanced their quality and production efficiency” is based on a strategy of successive harvesting and revegetation of rocket and spinach baby leaves in soilless culture for increased production efficiency and quality.

To sum up, this experiment can find an interest for the specialists in this field as well as it is applicable in soilless culture system the for their economical production. The following minor revisions are suggested.

Title: The title is fine and supports the arguments which have been addressed in this research, but include the names of both leafy vegetables (preferably scientific names) in the title.

  • Response: The title was amended as suggested.

Abstract:

Abstract is fine and appropriate to sum up the whole study.

Keywords: Keywords are appropriate

Introduction

Introduction is good and well elaborated the scope of study. 

Materials and Methods

Methodology of the proposed study is fine and well designed, but it will be more convenient if provide pictures of replantation especially revegetation or describe the intensity of harvesting to revegetate.

  • Response, L109-114: A description about harvesting practice was included, as suggested.

Results and Discussion

The results are described properly and supported with references but the minor changes are required as follow;

Line 136-146: These lines should be the part of Introduction section instead of Results and Discussions.

  • Response, L156-166: Appropriate parts of this paragraph were moved to the introduction section (L76-84) as suggested.

Line 155-160: Use the words “first, second and third etc.” instead of 1st, 2nd and 3rd etc.

  • Response: First, second and third etc. replaced 1st, 2nd and 3rd etc. throughout the manuscript, as suggested.

Provide the pictures for each harvest in supplementary files

  • Response: We would like to include pictures from our experiments but unfortunately we had a problem with the computer and the files were lost.

References

No reference is found unformatted.

Good Luck!

Reviewer 3 Report

The article is interesting and brings new knowledge in the field of hydroponic cultivation of nutritional plants. The article has great practical potential. Leafy vegetables are especially valuable because they provide many phytochemicals necessary for the proper functioning of the body. The more that thanks to such crops they can be available throughout the season.

Concerns are raised by the wide variation in water temperature between 6 and 15 ° C. Such values for the environment, which in this type of cultivation is a substitute for the substrate, seem too high, even for spinach and arugula, which are plants resistant to low temperatures. Did these fluctuations not affect the dynamics of plant growth?

Another question concerns the qualitative and quantitative composition of the medium. The authors are asked to clarify this aspect, which is important in this type of cultivation. Moreover, wouldn't it be worth using modern solutions that were created after 1950. After all, the cited solution is already 70 years old. Since then, some technical aspects and solutions have changed, such as the types of media, the type and form of nutrients.

Authors are requested to amend the abstract so that the necessary benefits of this article can be demonstrated in practice.

Author Response

The article is interesting and brings new knowledge in the field of hydroponic cultivation of nutritional plants. The article has great practical potential. Leafy vegetables are especially valuable because they provide many phytochemicals necessary for the proper functioning of the body. The more that thanks to such crops they can be available throughout the season.

Concerns are raised by the wide variation in water temperature between 6 and 15 ° C. Such values for the environment, which in this type of cultivation is a substitute for the substrate, seem too high, even for spinach and arugula, which are plants resistant to low temperatures. Did these fluctuations not affect the dynamics of plant growth?

  • Response, L105-108: The following phrase was included in the manuscript: Rocket and lettuce baby leaves were reported to have optimum production at 20 °C root zone temperature. Therefore, 6-15 °C fall within the favorable root zone temperatures, while they are typical for the experiment’s period of cultivation.

Another question concerns the qualitative and quantitative composition of the medium. The authors are asked to clarify this aspect, which is important in this type of cultivation. Moreover, wouldn't it be worth using modern solutions that were created after 1950. After all, the cited solution is already 70 years old. Since then, some technical aspects and solutions have changed, such as the types of media, the type and form of nutrients.

  • Response: Even though Hoagland solution is 70 years old, it still commonly used in similar experiments and plant production systems. We include a few recent articles where authors implemented the same nutrient solution:

Sudiarto et al., 2019: https://doi.org/10.1016/j.jenvman.2018.10.070

Zhou et al., 2019: https://doi.org/10.1016/j.envpol.2018.11.096

Si et al., 2020: https://link.springer.com/article/10.1007/s10452-020-09751-3

Authors are requested to amend the abstract so that the necessary benefits of this article can be demonstrated in practice.

  • Response: The abstract was amended as suggested.

Reviewer 4 Report

In the reviewed paper entitled 'Strategic successive harvesting of baby leaf vegetables enhanced their quality and production efficiency' is really hard to assess the benefits of successive harvesting. The reason is a lack of proper controls. I understand that control fluctuational changes were caused by external conditions which were not controlled, but because of this, the proper assessment was impossible. In my opinion, the experiment was not properly designed. First - the number of controls should be enlarged, second - the experiment was done only with two cultivars representing two species, so you can not make a statement about all species reaction - this is only the speculation about qualitative and quantitative positive changes.

The idea of successive harvesting is not only economically, but also ecologically very attractive nevertheless would be better to:
- take 3 or more distinct cultivars with different pedigrees representing only one of the analysed species
- conduct the experiment in controlled light conditions 
- to plan precisely control sowing to synchronize harvesting of controls and successive harvesting
- develop the method of controls reproducibility

I prefer not to join the results and discussion paragraphs as it harnesses finding the results obtained in the presented experiment.

I recommend reanalyse the results once again, maybe some of them could be used in other way and could be a basis of the next manuscript.

Author Response

In the reviewed paper entitled 'Strategic successive harvesting of baby leaf vegetables enhanced their quality and production efficiency' is really hard to assess the benefits of successive harvesting. The reason is a lack of proper controls. I understand that control fluctuational changes were caused by external conditions which were not controlled, but because of this, the proper assessment was impossible. In my opinion, the experiment was not properly designed. First - the number of controls should be enlarged, second - the experiment was done only with two cultivars representing two species, so you can not make a statement about all species reaction - this is only the speculation about qualitative and quantitative positive changes.

  • Response: Rocket and lettuce baby leaves were reported to have optimum production at 20 °C root zone temperature. Therefore, 6-15 °C fall within the favorable root zone temperatures, while they are typical for the experiment’s period of cultivation. It is now specified in L105-108.

It was not our intention to make a statement about the reaction of all species. In the conclusions’ section (L340-342) it is stated that “successive harvesting and revegetation is suggested for increased production efficiency and quality of rocket and spinach baby leaves”.

The idea of successive harvesting is not only economically, but also ecologically very attractive nevertheless would be better to:

- take 3 or more distinct cultivars with different pedigrees representing only one of the analysed species

- conduct the experiment in controlled light conditions

- to plan precisely control sowing to synchronize harvesting of controls and successive harvesting

- develop the method of controls reproducibility

  • Response: - Endless cultivars may be tested with regard to their qualitative and quantitative traits. However, one aim of our study was to investigate the specific species which are particularly popular throughout the world.

- Moreover, another aim of our study was to test successive revegetation and harvesting in greenhouse conditions where light cannot be fully controlled. Plant factories with artificial lighting (PFAL) are increasing in popularity but are not largely established in warm climates such as the Mediterranean region.

- In this experiment, sowings were precisely scheduled and the control tanks were implemented for the whole experimental period. However, plants in the control tanks needed more time to reach the harvesting stage, while plants in the revegetation tanks already had a well-developed root system which accelerated leaves’ development after cutting. Therefore, in a period of about 2 months, control and revegetated tanks led to 3 and 5 cuts, respectively.

- Baby leaves were harvested when they reached appropriate size (L109). In order for the control tank to be immediately used with plants of appropriate age, three successive sowings were done two days apart. This is now included in the manuscript in L112-114.

I prefer not to join the results and discussion paragraphs as it harnesses finding the results obtained in the presented experiment.

I recommend reanalyse the results once again, maybe some of them could be used in other way and could be a basis of the next manuscript.

Round 2

Reviewer 4 Report

I still believe that the proper performance of this analysis requires obtaining the appropriate reference - control.

Obtaining specific results on individual varieties - cultivars of a given species does not guarantee the same response from other varieties.

My final decision is dictated only by the positive assessment of the other reviewers. In my opinion, the accusations presented by me in the previous review disqualify this work.